# Assessment of Antimicrobic, Antivirotic and Cytotoxic Potential of Alginate Beads Cross-Linked by Bivalent Ions for Vaginal Administration

**DOI:** 10.3390/pharmaceutics13020165

**Published:** 2021-01-27

**Authors:** Miroslava Pavelková, Jakub Vysloužil, Kateřina Kubová, Sylvie Pavloková, Dobromila Molinková, Vladimír Celer, Alena Pechová, Josef Mašek, David Vetchý

**Affiliations:** 1Department of Pharmaceutical Technology, Faculty of Pharmacy, Masaryk University, Palackého 1, 612 00 Brno, Czech Republic; pavelkovam@pharm.muni.cz (M.P.); kubovak@pharm.muni.cz (K.K.); pavlokovas@pharm.muni.cz (S.P.); vetchyd@pharm.muni.cz (D.V.); 2Department of Infectious Diseases and Microbiology, Faculty of Veterinary Medicine, University of Veterinary and Pharmaceutical Sciences Brno, Palackého 1, 612 42 Brno, Czech Republic; molinkovad@vfu.cz (D.M.); celerv@vfu.cz (V.C.); 3Department of Animal Breeding, Animal Nutrition and Biochemistry, Faculty of Veterinary Hygiene and Ecology, University of Veterinary and Pharmaceutical Sciences Brno, Palackého 1, 612 42 Brno, Czech Republic; pechovaa@vfu.cz; 4Department of Pharmacology and Toxicology, Veterinary Research Institute, Hudcova 70, 621 00 Brno, Czech Republic; masek@vri.cz

**Keywords:** ionic gelation, alginate, cross-linking, bivalent ions, cytotoxicity, antimicrobial activity, antiviral activity

## Abstract

Antimicrobial agent abuse poses a serious threat for future pharmacotherapy, including vaginal administration. The solution can be found in simple polymeric systems with inherent antimicrobial properties without the need to incorporate drugs, for instance alginate beads cross-linked by bivalent ions. The main goal of the presented study was to provide improvement on the well-documented cytotoxicity of Cu^2+^ cross-linked alginate. Alginate beads were prepared by external ionotropic gelation by cross-linking with Cu^2+^, Ca^2+^ and Zn^2+^ ions, separately and in mixtures. Morphological properties, swelling capacity, ion release and efficacy against the most common vaginal pathogens (*C. albicans*, *E. coli*, *E. faecalis* and virus strain—human herpesvirus type 1) were evaluated. The prepared particles (particle size 1455.68 ± 18.71–1756.31 ± 16.58 µm) had very good sphericity (0.86 ± 0.04–0.97 ± 0.06). In mixture samples, Cu^2+^ hampered second ion loading, and was also released incompletely (18.75–44.8%) compared to the single ion Cu^2+^ sample (71.4%). Efficacy against the selected pathogens was confirmed in almost all samples. Although anticipating otherwise, ion mixture samples did not show betterment over a Cu^2+^ cross-linked sample in cytotoxicity–pathogen efficacy relation. However, the desired improvement was found in a single ion Zn^2+^ sample whose minimal inhibition concentrations against the pathogens (0.6–6.12 mM) were close to, or in the same mathematical order as, its toxic concentration of 50 (1.891 mM). In summary, these findings combined with alginate’s biocompatibility and biodegradability give the combination solid potential in antimicrobial use.

## 1. Introduction

Antimicrobial resistance is a serious problem for present and future pharmacotherapy of infectious diseases [1,2]. The issue could be partially overcome by using simple polymeric systems with inherent antimicrobial properties without needing actual drugs to be incorporated. Several new conceptions have been presented recently [3,4,5]. One of these approaches could be the use of alginate particles cross-linked with bivalent metal ions [6].

Alginates are widely used in the medicinal and pharmaceutical field. They have a wide range of favorable properties, such as biocompatibility, biodegradability, bioadhesivity [7,8] and structural similarity with the extracellular matrix, making them a biomaterial of choice in the area of wound dressings, tissue engineering and regeneration [9,10]. The formulation of the controlled release dosage forms [11] and possibility for protein drugs, probiotics, and other live cell encapsulation [12,13,14] are other areas of their numerous applications. Of the high number of various dosage forms based on alginates published regularly, cross-linked alginate particles possess a particulate character, which allows easy dose dispersal in the administration site, lowering local adverse effects, the possibility to use dispensing tools when administering to body cavities and are easily adjustable for controlled release by means of pharmaceutical technology [15]. Alginate itself enhances these benefits due to its previously recorded mucoadhesive effect, increasing contact time and thus lowering dose frequency [6].

From the chemical point of view, these polysaccharides are linear copolymers with different composition and sequence of β-D-mannuronic acid (M) residues and α-L-guluronic acid (G) linked by (1→4) glycosidic bonds, creating three basic building blocks [16,17]. Their most important physicochemical property is the ability to form gels [18,19]. Ionic gel formation depends mainly on the alginate affinity to the cross-linking ions, which decreases in series: Pb > Cu > Cd> Ba > Sr > Ca > Co, Ni/Zn > Mn [20,21]. Whereas the use of Ba^2+^, Sr^2+^, and Pb^2+^ is limited due to their potentially toxic effect on the human organism [22,23,24], the calcium(II) ion is the most used due to it having the safest profile and a solid cross-linking ability, being a golden standard for external ionic gelation [25]. Only few papers dealing with cross-linking via ion mixture have been published [26], even though it could provide some advantages including the reduction in single cation weaknesses regarding both technological and biological aspects. Whereas the calcium (II) ion is generally accepted as the most physiological ion for external ionic gelation, some of the polyvalent ions in the affinity series, such as Cu^2+^ or Zn^2+^, possess their own pharmacological effect, which offers the possibility of dosage form formulation without the need for any other drug encapsulation. While human tissue sensitivity to copper is low, the sensitivity of many microorganisms is extreme [27], including *Escherichia coli*, *Staphylococcus epidermidis*, vancomycin-resistant enterococci, methicillin resistant *Staphylococcus aureus*, yeasts such as *Candida albicans* or even viruses such as bacteriophages, herpes simplex virus, poliovirus, hepatitis A virus, and HIV [27,28,29,30,31]. Zinc has a similar antimicrobial spectrum as copper. The effectiveness against *Staphylococcus aureus* (including methicillin-resistant strains), *Bacillus subtilis*, *Enterococcus faecalis*, *Staphylococcus epidermidis*, *Streptococcus pyogenes*, *Streptococcus mutans*, *Pseudomonas aeruginosa* or *E. coli* [32,33,34,35,36,37,38], as well as some fungi [32,39,40] has been proven. The direct antiviral effect against various types of viruses, including human rhinovirus, herpes simplex virus, HIV, hepatitis C virus, respiratory syncytial virus and vaccinia virus, has also been confirmed [41].

The aim of this paper was to prepare alginate-based mucoadhesive cross-linked beads suitable for use against the most common vaginal infections. In previous experiments, significant Cu^2+^ cytotoxic effect was demonstrated [6], therefore in addition to Cu^2+^, physiologically more acceptable Ca^2+^ and Zn^2+^ ions were used for networking, either separately or in mixtures. To our knowledge no paper has dealt with ion combination cross-linked alginate particles and their biological activity profile. The main goal was to determine if it is possible to decrease the severe Cu^2+^ cytotoxic effect while maintaining a solid antimicrobial activity. With respect to each ion’s determined cytotoxic concentration, antimicrobial and antiviral efficacy against the most common vaginal pathogens were assessed. Comparing cytotoxicity and antimicrobial activity can indicate the possibility of a potential use and is often not addressed in many studies. Several parameters, such as yield, morphological properties, swelling capacity and the release of individual ions in the environment simulating inflammation were also evaluated.

## 2. Materials and Methods

### 2.1. Materials

Sodium alginate (NaALG) of medium viscosity grade (5.0–40.0 cps, for 1% dispersion in purified water) with G/M units’ ratio 1:1 and previously characterized primary structure and molecular weight of 94,000–99,000 [42] was used as a polymer carrier (Sigma Aldrich, USA). Aqueous solutions of CuCl_2_, ZnCl_2_ and CaCl_2_ were used as hardening agents (Penta, Czech Republic). Calibration solutions were prepared using a dilution of 1000 mg/L stock copper(II), zinc(II) and calcium(II) reference solvents (Analytika, Czech Republic). Deionized water with a resistivity of 18 MΩ was used for all necessary dilutions. Bacterial strains—*Escherichia coli* (*E. coli*) (CCM 4517) and *Enterococcus faecalis* (*E. faecalis*) (field strain), the yeast strain—*Candida albicans* (*C. albicans*) (CCM 8186), the virus strain—human herpesvirus type 1 (HHV–1) (ATCC VR-539), and Vero cell line (ATCC CCL-81) in DMEM medium supplemented with 10% fetal bovine serum (FBS) were used for microbiological evaluation.

### 2.2. Bead Preparation

The individual bead samples prepared from 120 g of NaALG dispersion were obtained through external ionic gelation [25]. Homogenized dispersion of NaALG (6%) was prepared by gradually swelling and dissolving NaALG in purified water for 15 min at room temperature, and subsequent homogenization using the Ultra-Turrax (T25 basic, IKA-Werke, Germany) at 13,000 rpm for 5 min. Air bubble elimination was ensured by an ultrasonic bath (Bandelin Sonorex RK 106, Germany) for 30 min. The homogenized dispersion was extruded through a 0.7 mm diameter needle at a flow rate 2.0 mL/min, into 50 mL of 1.0 M hardening solution (CuCl_2_, ZnCl_2_, CaCl_2_ or their 0.5 M:0.5 M mixtures). Concentrations were selected on the basis of our previous pre-formulation studies [6,42]. The distance between the edge of the needle and the solution’s surface was adjusted to 7.0 cm. Formed beads were immediately cured for 60 min, then thoroughly washed with purified water and dried in a cabinet drier (HORO-048B, Dr. Hofmann GmbH, Germany) at 25 °C for 24 h. The designation of the samples corresponded with the type of cross-linking ions (see Table 1).

### 2.3. Evaluation of Morphological Parameters

Optical microscope analysis: A NIKON SMZ 1500 stereoscopic microscope (Nikon, Japan) with a72AUC02 USB camera (The Imaging Source, Germany) was used to evaluate the equivalent diameter (ED) and sphericity factor (SF) using 15-fold magnification. A total of 200 beads were randomly chosen for each sample and processed using the NIS Elements AR 4.0 software (Nikon, Japan). ED and SF were calculated and expressed as an arithmetic mean with standard deviation (SD) according to the Equations (1) and (2), respectively [43,44].
(1)$$\mathrm{ED}=\;\sqrt{\frac{4A}{\pi }}\;\left[\mathrm{mm}\right]$$
(2)$$\mathrm{SF}\;=\;\frac{4\pi A}{p^{2}}$$where A is the area of the bead in square millimeter and p is the perimeter of the bead in millimeters.

Scanning electron microscopy (SEM): Scanning electron microscope (SEM; MIRA3, Tescan Orsay Holding, Czech Republic) equipped with a secondary electron detector (SED) was used to observe surface morphology. Samples fixed onto specimen stub with carbon conductive adhesive tape (Agar Scientific, United Kingdom) were sputtered by Au under argon atmosphere (Q150R ES Rotary-Pumped Sputter Coater/Carbon Coater, Quorum Technologies, United Kingdom) for charging artefact elimination. SEM images were obtained at the 3 kV accelerating voltage.

### 2.4. Swelling Capacity

Determining the swelling capacity was performed according to the previously reported method [45] in a phosphate buffer of pH 6.0, prepared per European Pharmacopoeia (Ph. Eur.) 10th Ed. (6.8 g of sodium dihydrogen phosphate R in 1000 mL of water R, pH adjusted with concentrated sodium hydroxide solution) [46], to mimic the pH value present during vaginal inflammation [47]. From each sample, 100 mg were put into fine mesh baskets, immersed in 45 mL of the phosphate buffer, withdrawn at 0.5, 1, 2, 3, 4 and 6 h, dried properly and weighed. The calculation carried out according to the following Equation (3):(3)$$S_{\mathrm{SW}}\;=\;\left(\frac{W_{t}{\;-\;W}_{0}}{W_{0}}\right)\times \;100\;\left[\%\right]$$
where S_SW_ is the swelling capacity (weight gain percentage), W_t_ is the sample weight at the given time and W_0_ is the initial weight [45]. Each sample was measured in triplicate and the results were expressed as mean values with SD.

### 2.5. Ion Content

The ion content in the beads was determined by atomic absorption spectrometry. The samples (20–40 mg) were digested in a TFM digestion vessel using 6 mL of concentrated nitric acid (65% *v*/*v*) and 2 mL of hydrogen peroxide (30% *v*/*v*). Mineralization was performed at 220 °C in Ethos SEL Microwave Labstation (Milestone, Italy) for 35 min (a 15 min steady increase—20 min holding maximum temperature), applying a maximum power of 1000 W. After cooling, each resulting solution was transferred to a 50 mL glass flask and filled to the mark with deionized water. The samples were diluted with deionized water (to determine Cu and Zn) and ionized buffer (to determine Ca) prior to further analysis. Copper and zinc content were measured using air-acetylene flame atomization, calcium content was determined by nitrous oxide–acetylene flame atomization (contrAA 700, Analytik Jena, Germany). The samples were measured in triplicate; the results were processed and expressed as mean value with SD (Aspect CS software, version 2.1.).

### 2.6. Release Profiles of Individual Ions from Beads

The bead solutions (10% *w*/*v*) were prepared to determine the ion release characteristics. Of each sample, 0.5 g was put into a 50 mL glass vessel (with a bottom diameter 4 cm), 4.5 mL of pH 6.0 phosphate buffer was added, and the vessels were placed into a cabinet drier (HORO-048B, Dr. Hofmann GmbH, Germany) set at 37 °C to simulate the human body temperature during the alternative dissolution test. At the 0.5, 1, 2, and 6 h, the vessels were placed on a horizontal shaker (HS 250 basic, IKA Labortechnik, IKA Werk, Germany) for 1 min at 120 reciprocating movement/min, the samples were filtered (0.22 μm membrane filter) and the ion content was determined using atomic absorption spectroscopy (see Section 2.5). All samples were measured in triplicate and expressed as mean value with SD.

### 2.7. Biological Effects

For all tests, a suitable amount of alginate beads was suspended for 60 min at room temperature in the appropriate growth medium to prepare a 10% suspension. The undissolved beads were separated via centrifugation, discarded, and then the concentrations of the released ions in the leachates were determined using the atomic absorption spectroscopy.

*Determining cytotoxicity:* The cytotoxicity was tested on the Vero cell line. Vero cells (105/mL) were suspended in a DMEM medium (with 10% FBS), and 1000 μL of cell suspension per well was seeded into a 24-well plate (Nunc). Serial tenfold dilutions of individual sample leachates in the same medium (1000 μL) were added into a 24-h-old monolayer of Vero cells after removing the seeding medium, nine wells per dilution (10-1 to 10-9). The plates were then put into a CO_2_ incubator for 48 h at 37 °C. The number of viable cells after 24 and 48 h of incubation was counted in the Bürker chamber using methylene blue staining (10 μL of 0.5% methylene blue aqueous solution was mixed with the same volume of trypsinated cells, staining dead cells blue). The viable cells were compared with the number of cells in control wells (containing untreated Vero cells). Cell viability for each of ion dilutions was expressed as a percentage of viable cells according to the following Equation (4):(4)cell viability = A/B*100 [%] 
where A represents the quantity of viable cells in the wells with ions and B is the number of cells from the control wells. The TC50 (50% toxic concentration) value was determined for 24 and 48 h time points as the first dilution of ion leachate that caused a 50% reduction in viable cells [48]. The cell viability evaluation at 48 h was used for in vitro assessment of cellular processes as consequences of interactions between ions and cellular structures. On the other hand, the ion dilution where the cell number was equal to the control (untreated Vero cells) well was selected as a non-cytotoxic concentration.

*Antimicrobial activity:* The antimicrobial activity of all samples against yeast *Candida albicans*, Gram-negative bacteria *Escherichia coli* and Gram-positive bacteria *Enterococcus faecalis* (as the most common causes of the vaginal infections) was evaluated [49]. The bacterial cultures were prepared by suspending an overnight culture of *E. coli* and *E. faecalis* in fresh Luria broth medium and left to grow to OD600 0.5 at 37 °C and shaking speed of 250 rpm. *C. albicans* was prepared in Yeast Nitrogen base medium with Ammonium Sulphate under the same conditions. Twofold dilutions of the 10% bead suspension leachates were titrated into a microtiter plate (100 μL/well in triplicate) and then 106 cells (*E. coli*, *E. faecalis* or *C. albicans*) were added to each well. These mixtures were incubated for 20 h at 37 °C. To monitor cell growth, optical density was measured spectrophotometrically (OD600) following the incubation. Individual ions’ antimicrobial activity was expressed as the minimal inhibition concentration (MIC). This concentration is defined as the ion concentration in a well showing at least a fourfold reduction in OD600 absorbance compared to each subsequent well [6]. The mean of the three wells was calculated to evaluate the MIC. Wells containing bacteria or yeast without ion inhibition were included in all tested plates as controls.

*Antiviral activity*: A plaque reduction assay was used to determine the number of viable HHV-1 particles after exposure to the ions and their mixtures. Determining the antiviral activity was based on monitoring the number reduction in the cytopathogenic effects (CPEs). The preventive and therapeutic effects of these ions in contact with the HHV-1 virus were tested, which means before and after HHV-1 entered the cells. First, it was necessary to determine the optimal HHV-1 concentration as well as the optimal (non-cytotoxic) concentration of ions in the individual leachates. The virus stock was diluted tenfold in DMEM (with 2% FBS). One milliliter of the virus solutions was added into 24-well microtiter plates (Nunc) containing an 80% confluent monolayer of the Vero cells, three wells per dilution. After 24 h incubation, the CPEs were counted using crystal violet staining and the quantity of plaque-forming units in 1 mL of stock was calculated as an average of the same three wells. Determining the optimal (non-cytotoxic) concentration for the ions used was evaluated (see Determination of Cytotoxicity). For each ion or combination, one 24-well plate (Nunc) was used. Six wells with an 80% monolayer of Vero cells were infected with the virus in optimal concentration and the other six wells were covered with the ionic solution in non-toxic concentration. The plates were incubated at 37 °C with 5% CO2 for one hour and then the relevant quantity of ions was pipetted into the wells with HHV-1 and vice versa. Untreated Vero cells, cells with only the ions and cells with the HHV-1 virus were included as the controls. After that, all the plates were incubated at 37.5 °C with 5% CO2 for 48 h. Then, the cells were stained by crystal violet and the number of CPEs was counted.

All results obtained during the cytotoxicity and antimicrobial activity evaluation were subjected to the statistical analysis of significance using ANOVA (R software).

## 3. Results and Discussion

### 3.1. The Evaluation of Morphological Parameters

The particle size obtained via microscope analysis ranged between 1455.68 ± 18.71 μm and 1756.31 ± 16.58 μm (see Table 2), interval boundaries belonging to the samples ALG_Cu and ALG_Zn:Ca, respectively. Within the samples cross-linked using a single ion, the smallest beads were formed using Cu^2+^ ions (1455.68 ± 18.71 μm) and the largest were prepared using Zn^2+^ ions (1688.44 ± 12.45). These results were expected, as the Cu^2+^ ions, which have the strongest affinity, are non-specifically bound to the G- and M- units [50] and, therefore, a denser network was created than with Zn^2+^ and Ca^2+^ ions. Conversely, Zn^2+^ ions exhibit the weakest affinity to the alginate and their binding affinity to M-units is minimal [42], which resulted in an expectedly larger particle size [25,51,52]. The samples cross-linked by ionic mixture reflected the same pattern. Mixture beads containing Cu^2+^ ions exhibited smaller ED values, whereas the beads cross-linked by a mixture of Zn^2+^ and Ca^2+^ ions showed higher ED. The similar dependency was also evident in the sphericity assessment; the samples with Cu^2+^ had higher sphericity values. However, the average sphericity factor (SF) was satisfactory for all samples, i.e., greater than 0.8 (which is the limit value for pellets sphericity) [53] (see Table 2).

Different binding preferences (to G- and M-units) together with a different degree of ion affinity to the alginate also had an important impact on the morphological character of the bead surface, which is obvious from the SEM images (Figure 1 and Figure 2). The surface of beads cross-linked by a single ion (Figure 1) show a significantly greater number of deep cracks and roughness than the beads cross-linked with ion mixtures. The most compact and rigid surface, and the most spherical shape is obvious in the ALG_Cu sample (Figure 1A). The whole bead image shows many fine, thread-like cracks on the regularly cross-linked surface. However, the cracks are not as markedly deep as in ALG_Zn and ALG_Ca samples. The ALG_Zn surface detail image (Figure 1B) shows the less regularly cross-linked structure with less numerous but deeper cracks. The ALG_Ca beads (Figure 1C) exhibit a very regularly cross-linked structure. Their cracks are much less numerous but very pronounced and they seem to reach deeper structures than previous samples.

Looking at the SEM bead images cross-linked by ion mixtures (Figure 2), more regular structures are visible with a more compact surface than cross-linking by a single ion. Specifically, the ALG_Cu:Zn sample exhibits a considerably higher degree of cross-linking leading to the most compact surface from all samples being formed. It does not contain apparent cracks; only in a more detailed resolution are the mild furrows noticeable. The other two samples show a regularly cross-linked surface which can be declared as the smoothest of all the samples; however, a certain number of cracks are evident. The main factors influencing the final character of the bead surface (their cracks, inequalities and roughness) include the type of cross-linking ions used, their affinity to alginate and also concurrently suitable temperature and drying time [54].

### 3.2. Ion Content

The overview of the individual ion contents acquired by atomic absorption spectrometry is shown in the Table 3. The ALG_Zn sample contained the highest ion amount (223.8 ± 21.3 g/kg), followed by the ALG_Cu sample with 205.3 ± 25.1 g/kg of copper. With respect to the standard deviations, the difference between them was negligible. The explanation of the high copper content inside the beads could be seen as a stronger copper alginate affinity leading to the almost immediate replacement of sodium ions and formation of a very dense electrostatic bond network between Cu^2+^ and COO^-^, especially in the outer parts of the beads [55]. Nevertheless, the highest ion content in the ALG_Zn was a surprise. The explanation for this could be as follows: the size of Zn^2+^ and Cu^2+^ ions is almost the same (Cu^2+^ 73 pm, Zn^2+^ 74 pm) and from this point of view, the ability to move among the alginate chains could be very similar. However, Zn^2+^, as an ion with a significantly lower alginate affinity [21], does not cross-link the bead surface so quickly, which enables free Zn^2+^ permeation into the bead core with no obstacle. In contrast, the smallest ion content, namely, 117.4 ± 15.9 g/kg, was detected in the ALG_Ca sample. The significantly larger ionic radius of Ca^2+^ (100 pm) can hinder the permeation, providing an explanation for the low ion content.

The samples cross-linked by an ion combination with Cu^2+^ contained significantly more Cu^2+^ ions than the second ion used. The markedly higher Cu^2+^ affinity to alginate together with their smaller radius played a crucial role in formulating the solid ring in the outer part of the bead. This created a strongly cross-linked barrier [56] that was probably able to prevent subsequent Zn^2+^ and Ca^2+^ penetration deeper into the bead structure. This could confirm the above-mentioned theory about the ion content differences in single ion samples. Moreover, the ion radius could also play a crucial role in the amount of the bound ions in the ALG_Zn:Ca sample. Zinc ions were present in a significantly higher amount (Zn^2+^ 123.4 ± 13.7 g/kg, Ca^2+^ 71.8 ± 5.1 g/kg), regardless of their low affinity to the alginate. The smaller Zn^2+^ ion size could result in the easier ion penetration in larger quantities into the bead structure during maturation.

### 3.3. Swelling Capacity

The swelling capacity of any hydrophilic polymer is a prerequisite to develop the previously demonstrated mucoadhesive effect [6], which permits extended residence time at the desired site of action without the need for frequent administration [57]. The swelling capacity evaluation was performed in pH 6.0 phosphate buffer, which simulated the inflamed vaginal environment [47]. The course of the individual samples’ swelling behavior is shown in Figure 3 and Figure 4. The samples containing Zn^2+^ and Ca^2+^ ions (either single or in a mixture) gradually swelled for 6 h. During the first 2 h, the ALG_Zn and ALG_Ca samples showed a relatively small water increase (55.1% and 89.53%), in the next 2 h the swelling capacity increased to more than double, and the final swelling values (after 6 h) grew up to 941.61% and 763.23%. The sample cross-linked by the mixture of these ions (ALG_Zn:Ca) exhibited a similar tendency to swell with an even higher final swelling value (1237%) [58]. Conversely, all beads cross-linked by Cu^2+^ ions (either alone or in mixture) swelled significantly less. During the first hour, the ALG_Cu sample reached the value 57.2% and subsequently the swelling rate increased only slightly; after 6 h, it was only 66.37%. The mixed beads containing Cu^2+^ showed a similar swelling course, with a slightly higher water increment (see Figure 4). Comparing the swelling charts of Figure 3 and Figure 4 shows that at the end of the first hour all swelling profiles are comparable, but while swelling increased greatly in the other samples, those cross-linked with Cu^2+^ swelled negligibly in the following hours. This fact can be explained by the strong Cu^2+^ affinity to alginate compared to Zn^2+^ and Ca^2+^. A more densely cross-linked structure and higher rigidity of beads containing copper presumably prevent water molecules gradually penetrating inwardly to the bead structures [56].

### 3.4. Release of Individual Ions

The ion release from the alginate beads was monitored after contact with the phosphate buffer (pH 6) to mimic pH during vaginal infection [47] and measured using atomic absorption spectrometry. The test served for sample comparison similarly to the classic dissolution test. In reality, slower release would be expected thanks to the limited vaginal fluid supply and longer dosage form residence time. Therefore, the test represents the ceiling of achievable values. During the first 30 min, massive immediate ion release was demonstrated (see Figure 5 and Figure 6). Subsequently, a lower amount was released for the rest of the test. In comparison with other single ion samples, the ALG_Ca beads showed relatively large quantities of the released ions (93.4% over 6 h). The largest Ca^2+^ radius and its average binding affinity were probably responsible for easy alginate chain disintegration during the dissolution test. The massive Ca^2+^ exchange for Na^+^ and H^+^ ions and the water penetration into the bead structure were following [59]. On the other hand, ALG_Zn and ALG_Cu samples released a similar number of ions (73.1% and 71.4%, respectively) after 6 h, but probably for different reasons. Zinc ions were not released completely, probably due to their presence in the deeper parts of the beads (where they accumulated thanks to their small radius). Incomplete Cu^2+^ ion release was probably caused by the strongest Cu^2+^ affinity to the alginate.

The samples cross-linked by an ion mixture showed significant differences in the amounts of individually released ions, particularly in the case of samples containing copper. The small amount of Cu^2+^ ions was released after 6 h from both samples (44.8% from the ALG_Cu:Zn and 18.75% from the ALG_Ca:Cu) compared to the Zn^2+^ (96.7%) and Ca^2+^ (68.4%) ions. Cu^2+^ creates a firm layer in the upper layers of the bead structure thanks to its strong affinity. Almost complete Zn^2+^ release can be explained by the low Zn^2+^ affinity to alginate, combined with its small ion diameter. On the other hand, Ca^2+^ affinity is stronger, the ion is bigger, and the release can probably be hampered by the Cu^2+^ barrier, resulting in a lower amount released than in the single ion Ca^2+^ sample. The different amount of Cu^2+^ released in the mixture and single ion samples can be explained by the different hardening solution concentration. It is very possible that the 1 M concentration used in the single ion sample actually oversaturated the offer of ALG binding sites, resulting in redundant Cu^2+^ encapsulated without binding, which could then be easily released. The ALG_Ca:Zn sample showed approximately the same amount of both released ions, namely, 90% of Ca^2+^ and 98.06% of Zn^2+^. Once again, this can be explained by of Ca^2+^ and Zn^2+^’s lower alginate affinity, resulting in more complete ion release. Moreover, the presence of Na^+^ in the dissolution medium also contributes to the easier exchange of these cross-linking ions [60].

### 3.5. Cytotoxicity Evaluation

The used cross-linking ions’ cytotoxicity evaluation (see Figure 7) was an essential part of this project due to the intended vaginal administration. The used medium volume mimicked the physiological conditions and the real amount of vaginal fluid [61]. The statistical analysis of significance was performed using ANOVA (R software). The ALG_Cu represented a starting point from our previous study [6]. The primary TC50 of Cu^2+^ was 1.953 mM. However, over 48 h, due to continuing ion interactions with cellular structures, this primary TC50 value was considerably reduced up to value 0.1953 mM. The obtained data were in line with the original observation (primary TC50 was 1.885 mM and decreased over time) [6]. These results are further supported by the literature including recent papers [62,63]. Thus, due to the increasing negative effect of Cu^2+^ ions on cell viability, their use as a cross-linking agent should be limited only for external and short-term administration. An opposite course in the development of a toxic effect on Vero cell culture was observed within the Zn^2+^ cytotoxicity evaluation. The primary TC50 value for Zn^2+^ (after 24 h) was 0.1891 mM. However, over 48 h, this primary TC50 of Zn^2+^ increased up to 1.891 mM, which shows a weakening of Zn^2+^ toxic influence on the cell viability. It can be concluded that after a period of the cellular process’ paralysis, the physiological reparative cell processes followed, leading to viable cell proliferation. Unexpectedly, the lowest TC50 was found for Ca^2+^ ions, specifically 0.1709 mM at 24 h. However, similar to the Zn^2+^ results, over 48 h, efforts to ensure cellular tissue viability probably led to reparative cellular processes, resulting in an increase in TC50 up to a value 17.09 mM. These findings could be explained based on interactions between ions, cellular structures, and cellular processes. After an initial sharp increase in intracellular calcium concentration, the proliferation stopped, and some cells died because of metabolic reasons. Even though Ca^2+^ is generally viewed as a safe ion for external ionic gelation, it is a major signaling molecule and its unbalanced higher concentrations in cells have a toxic effect on mitochondria and endoplasmic reticulum [64]. It is possible that Ca^2+^ released from the beads reduced the cell population by this mechanism. Nevertheless, surviving cells’ reparative processes (very much as in epithelial cells) led to the homeostatic calcium ratio adjustment and reactive huge cell proliferation [65]. In vivo, the initial toxicity would be also considerably lowered by dynamic fluid exchange, lower volumes and larger area, desquamation, etc. Therefore, if we presume that it is possible to monitor the cytotoxicity as a dynamic property for 48 h, based on the above-mentioned results, the calcium as a cross-linking agent for the alginate beads’ preparation can be deemed as relatively safe [66].

The experiment with ion mixture samples followed in order to assess possible reduction in Cu^2+^/Zn^2+^’s toxic effects when combined with another ion. The ALG_Cu:Zn sample showed the cytotoxic effect on Vero cells at the concentration of 0.3012 mM (Cu^2+^) and 0.3463 mM (Zn^2+^) at both time points, 24 and 48 h. These TC50 values were of the same mathematical order as TC50 in the individually tested ions [67], meaning that the lowering of the toxic potential of individual ions, so-called antagonistic effect, was not achieved. The situation was also similar in the combination of Zn^2+^ and Ca^2+^. The highest cytotoxic effect was found in the ALG_Ca:Cu sample. After 48 h, TC50 decreased even down to 5.998/4.345 mM. Gong et al. reported that including Ca^2+^ leads to a cytotoxicity decrease [39]; however, in our experiment, the assumption that an ion mixture (as a cross-linking agent) will lead to a significant reduction in toxic effects while maintaining the antimicrobial effect has not been confirmed. Nevertheless, the ion mixture results fall in line with single ion evaluation, where Ca^2+^ showed very high initial cytotoxicity. Combining it with heavy and persistent Cu^2+^’s cytotoxic effect thus clearly yields the most toxic effect achieved.

Nevertheless, for definite conclusions regarding cytotoxicity, further results including a biochemical description of the interaction of ions at the cellular level should be obtained. Although they are basically the body’s own ions, it is always necessary to consider the total amount released as well as the site of their effect. The stated cytotoxicity assessment methodology did not consider the presence of cervical mucus [68,69] and daily cyclic vaginal mucosa desquamation, which would undoubtedly contribute to mitigating the cytotoxic effects of the tested ions in the real vaginal environment.

### 3.6. Antimicrobial Activity

Antimicrobial activity in all samples was evaluated by MIC determination against the most common vaginal pathogens (*C. albicans*, *E. coli* and *E. faecalis*) [49]. The MIC is defined as the lowest ion’s concentration inhibiting visible growth of tested microorganisms [70]. The selected microorganisms were cultivated at 37 °C, which was the optimal temperature for bacterial growth as well as physiological body temperature. The results of the MIC evaluation can be seen in Figure 8, Figure 9 and Figure 10.

While the calcium ions did not exhibit an antimicrobial effect against any of the tested pathogens, Cu^2+^ and Zn^2+^’s antimicrobial activities were clearly demonstrated. The copper MICs for *E. coli* and *E. faecalis* were 5.94 mM and 2.95 mM, respectively. The MIC for *C. albicans* was higher (23.25) due to a relatively greater yeast resistance to the copper effect. Schwartz et al. reported an increase in resistance to excess copper and proposed that the increase is due to a combination of decreased copper uptake and an increase in copper chelation by metallothioneins [71]. The MIC of Zn^2+^ against *C. albicans* was 6.12 mM and against both, *E. coli* and *E. faecalis*, it was 3.02 mM. The moment we compare obtained data with ion cytotoxicity, it is evident that the beads cross-linked by Zn^2+^ represent a major improvement. Unlike the Cu^2+^ sample, the MICs and TC50 values in the ALG_Zn are in the same mathematical order, suggesting a better safety profile while keeping the antibacterial effect, with a superior effect against *C. albicans*. This finding is supported by the literature [39,72].

In ion mixture samples, particularly the ALG_Cu:Zn, a more powerful antimicrobial effect was achieved. The effects against yeast and bacteria manifested at lower individual ion concentrations. The synergistic antimicrobial effect of Cu^2+^ and Zn^2+^ ions was probably the reason for MICs reduction in both ions [67]. Samples containing Ca^2+^ showed slightly weaker effect, but were still comparable or even superior to single ion beads. The only exception was ALG_Zn:Ca’s inactivity against *E. faecalis*. However, comparison with the cytotoxicity results shows that the TC50 values of ion mixture samples lie in lower mathematical order than the antimicrobial concentrations, which could result in an adverse effect during eventual administration.

### 3.7. Antiviral Activity

The evaluation of inhibitive properties of bivalent ions against HHV-1 as a model virus can be seen in Table 4. The sample dilutions used corresponded with the cytotoxicity evaluation. The amounts of the cytopathogenic effects on the cell monolayer, stained by crystal violet, were the key factor in assessing the antiviral efficiency (see Figure 11). Both preventive (prior to the HHV-1 treatment of the Vero cells) and therapeutic (after the HHV-1 treatment of the Vero cells) effects were tested. As expected, the ALG_Ca sample showed the smallest antiviral efficacy; its inhibition potential was only 29–38%. Conversely, Cu^2+^ and Zn^2+^ ions exhibited a strong inhibitory potential. Especially, Cu^2+^ ions manifested as the most effective in inhibiting the formation of CPEs (as result of HHV-1 infection). Copper managed to inhibit CPE formations by up to 94% (when administered preventively) and up to 76% (when administered therapeutically). The principle of copper antiviral activity is the oxidative damage of the virus genetic material, specifically destructing the phosphodiester bonds in DNA chains [73]. As a result, the virus is essentially harmed with no chance to recover. Based on this mechanism of action, the copper efficacy against a broad spectrum of viruses can be expected. On the other hand, zinc ions’ antiviral effect (86% if administered preventive and 60% therapeutically) is likely to be targeted against a narrower spectrum of viruses. Their inhibitory effect against herpes viruses is described as an interaction with the glycoproteins on the outside of the virus [74], thus making the metal possibly less effective against other viruses lacking this structure. These results fall in line with studies carried out earlier, which showed that copper and zinc had a potent inhibitory function against some viruses [73,74,75,76,77].

The antiviral inhibitory effect of the ion mixtures was most likely ensured solely by the presence of the Cu^2+^ or Zn^2+^ ions. The ALG_Ca:Cu ion sample showed 92% antiviral effect (the ALG_Cu sample had 94% efficiency). In addition, Cu^2+^ concentration in both samples presenting an antiviral effect was similar (3.05 mg/mL in the ALG_Ca:Cu sample, 3.81 mg/mL in the ALG_Cu sample). A similar result can be seen when comparing ALG_Zn and ALG_Zn:Ca samples. Similar to the antimicrobial activity, the synergistic effect of the individual ion activity in the sample ALG_Cu:Zn was not demonstrated. In the preventive treatment, these ions inhibited the formation of 85% cytopathogenic effects. Based on information gathered from this experiment, no clear benefit in antiviral activity was found when comparing an ion mixture with a single ion. These findings are consistent with the literature [73,74,75,76,77].

Regarding the question of whether the ions are more efficient when introduced before or after the HHV-1 infection, it is obvious all samples are more efficient in preventive administration (before the introducing HHV-1). The explanation could be the fact that in this way the cells are able to maintain ion levels more effectively, granting some protection against the virus infection [64]. Moreover, spread of the HHV-1 virus within the human body occurs by travelling from cell to cell via the tight cell junctions [78], and therefore, the virus is not exposed to the hostile extracellular environment. This fact contributes to the decreasing inhibitory effect of the metal ions.

## 4. Conclusions

In our experiment, alginate beads cross-linked by either individual bivalent ions (Cu^2+^, Zn^2+,^ Ca^2+^) or by a mixture of two ions were prepared using the external ionotropic gelation. The efficacy against the most common vaginal pathogens (*C. albicans*, *E. coli*, *E. faecalis* and HHV-1) was confirmed in almost all samples. Although anticipating otherwise, the ion mixture cross-linked samples showed even higher cytotoxicity than the previously studied samples cross-linked with Cu^2+^. The best efficacy of ALG_Cu:Zn, expected due to the Cu^2+^/Zn^2+^ antimicrobial synergistic effect and cytotoxicity lowering the antagonistic effect, was not confirmed. Improvement was also not delivered by combination with Ca^2+^. Contrary to this, Ca^2+^ showed a surprisingly high initial cytotoxicity. This observation is notable, as in this field of expertise, Ca^2+^ ion is often deemed a safe cross-linking agent due to its physiological nature. This may be correct when the ion is used alone, however, in combination with another active ion or substance, the cytotoxic effects may be augmented and thus, to reveal such cases, we suggest performing cytotoxicity assessment.

An optimal candidate was finally found in a sample cross-linked with sole Zn^2+^ ions. The comparison of antimicrobial activity evaluation and cytotoxicity showed that the values are in the same mathematical order, meaning the formulation could keep antimicrobial effect without severe local adverse effects, further reduced by the dosage form’s particulate character. Using a suitable applicator, the dosage form would be ideal for vaginal administration to treat common local infections. Further studies will address an in vivo release simulation and development of application device.

In conclusion, possessing the mechanism of action that does not promote the resistance development, Zn^2+^ cross-linked alginate particles (and alginate dosage forms in general) could represent a suitable alternative to antibiotics, especially for local administration. On top of that, the resulting products are eco-friendlier, as the Zn^2+^ ion is naturally present in the environment. In small quantities, it should be easily incorporated into salts, and it would not create depos, not mentioning alginate as a carrier’s biocompatibility and biodegradability.

## Figures and Tables

**Figure 1 pharmaceutics-13-00165-f001:**
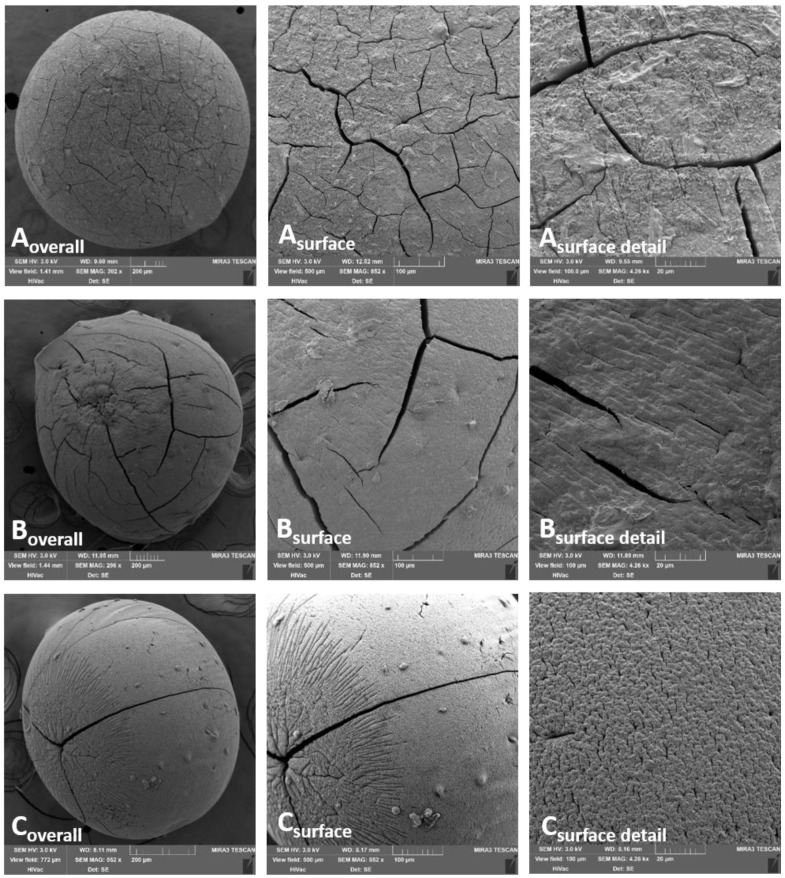
Alginate beads cross-linked by single bivalent ion, visualized using SEM technique: (**A**) ALG_Cu, (**B**) ALG_Zn; (**C**) ALG_Ca.

**Figure 2 pharmaceutics-13-00165-f002:**
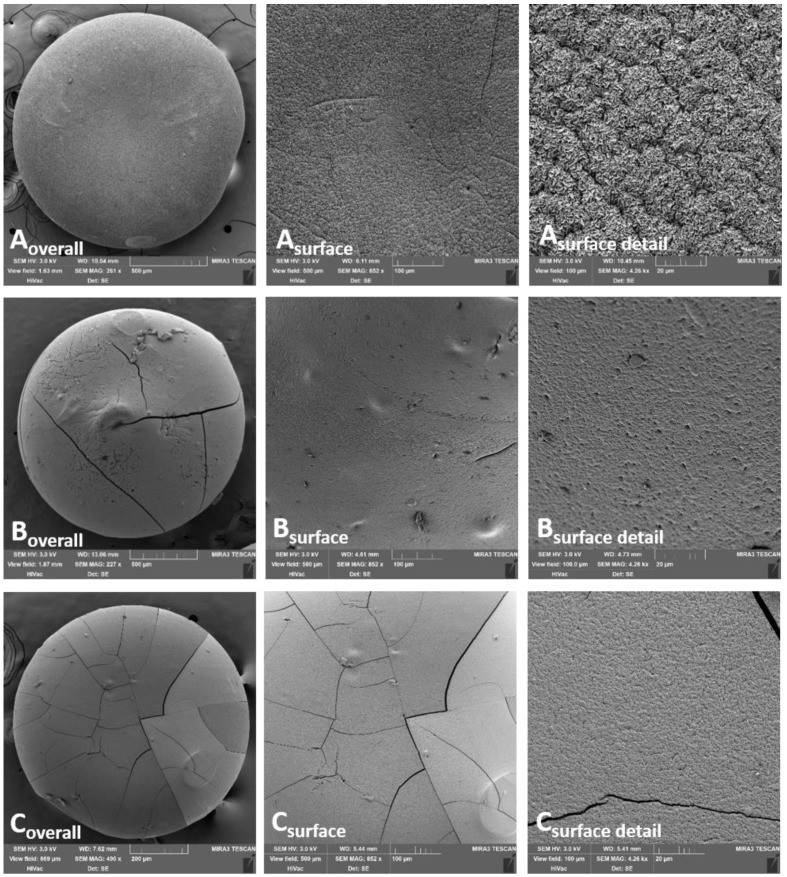
Alginate beads cross-linked by ion mixture, visualized using SEM technique: (**A**) ALG_Cu:Zn, (**B**) ALG_Zn:Ca; (**C**) ALG_Ca:Cu.

**Figure 3 pharmaceutics-13-00165-f003:**
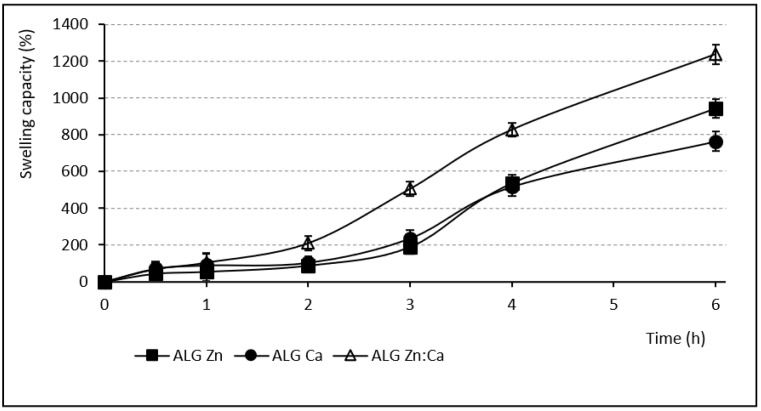
Swelling capacity of samples containing Zn^2+^ and Ca^2+^ ions (SDmax 61.2%, SDmin 20.2%).

**Figure 4 pharmaceutics-13-00165-f004:**
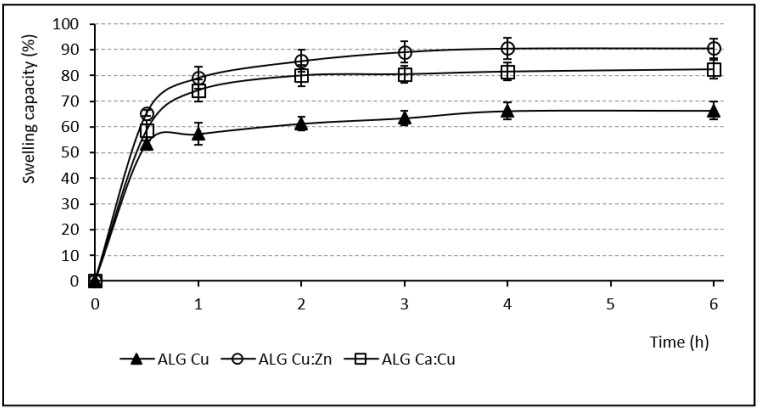
Swelling capacity of samples containing Cu^2+^ ions (SDmax 4.3%, SDmin 1.2%).

**Figure 5 pharmaceutics-13-00165-f005:**
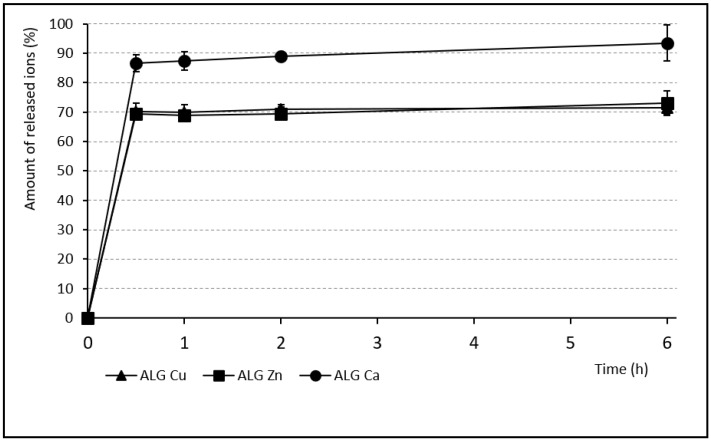
Release of ions from samples containing single ion (SD_max_ 6.1%, SD_min_ 2.2%).

**Figure 6 pharmaceutics-13-00165-f006:**
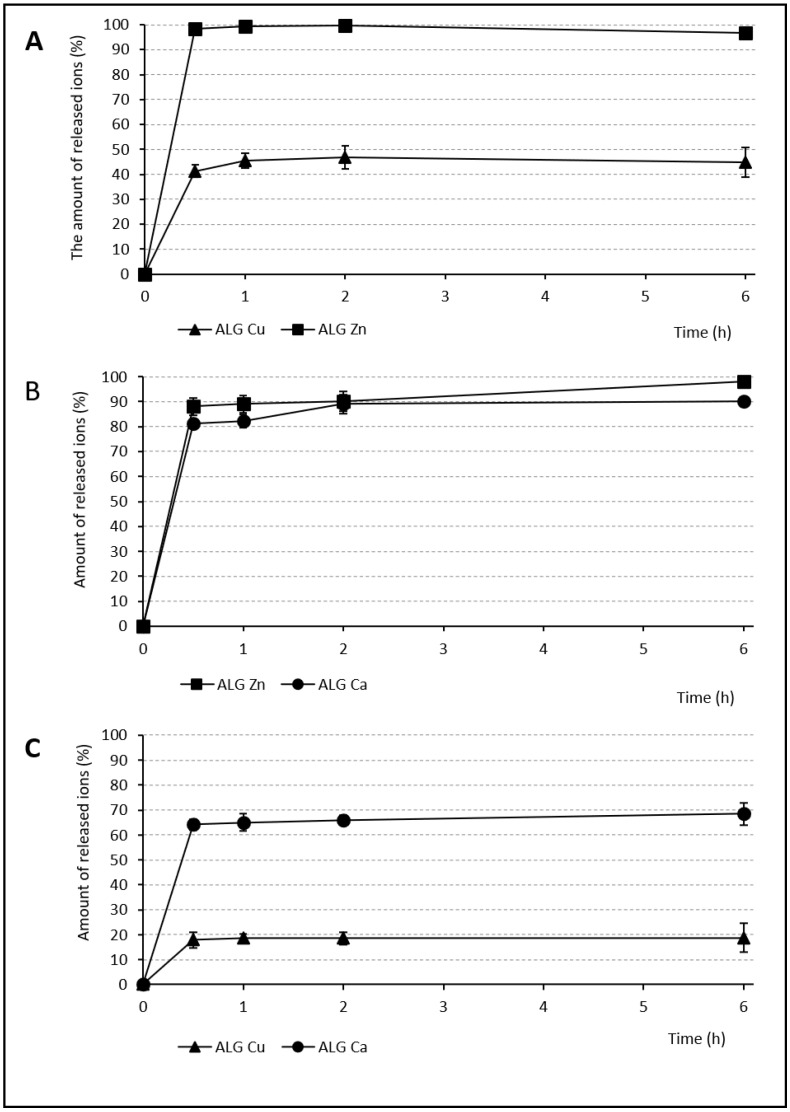
Release of ions from samples: (**A**) ALG_Cu:Zn, (**B**) ALG_Zn:Ca, (**C**) ALG_Ca:Cu (SDmax 5.9%, SDmin 0.2%).

**Figure 7 pharmaceutics-13-00165-f007:**
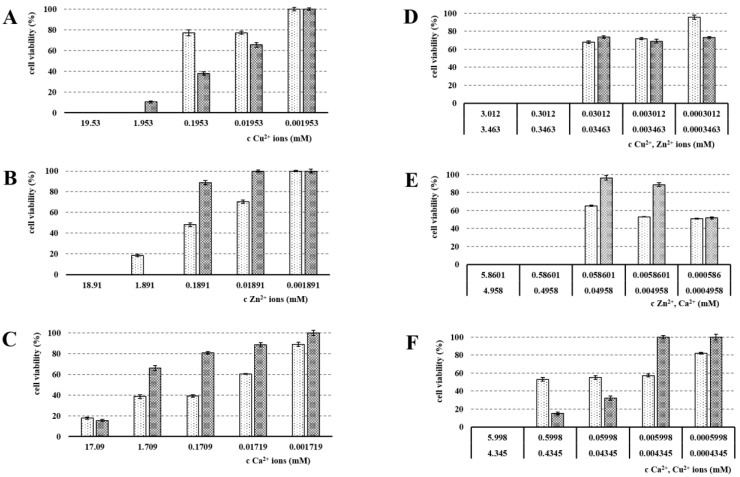
The viability of Vero cells (%) after treatment using different concentrations of Cu^2+^, Zn^2+^ a Ca^2+^ ions from (**A**) ALG_Cu, (**B**) ALG_Zn, (**C**) ALG_Ca, (**D**) ALG_Cu:Zn, (**E**) ALG_Zn:Ca, (**F**) ALG_Ca:Cu beads; 24 h (white), 48 h (grey), the statistically significant effect of ion concentration on viability of Vero cells after 24 and 48 h was proved by ANOVA (*p* < 0.001 in all cases).

**Figure 8 pharmaceutics-13-00165-f008:**
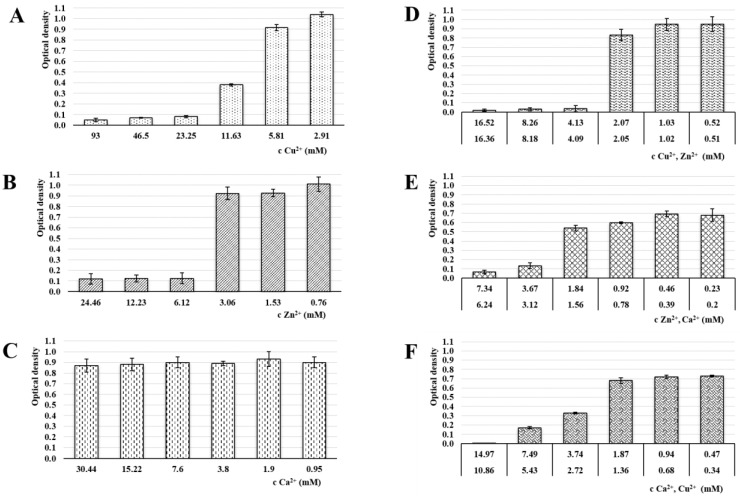
Optical density of *C. albicans* in media of individual samples: (**A**) ALG_Cu, (**B**) ALG_Zn, (**C**) ALG_Ca, (**D**) ALG_Cu:Zn, (**E**) ALG_ Zn:Ca and (**F**) ALG_Ca:Cu ions following 24 hrs incubation. The statistically significant effect of ion concentration on optical density of *C. albicans* in media was proved by ANOVA (*p* < 0.05 in all cases with the exception of ion concentration effect for Ca^2+^
*p* = 0.703).

**Figure 9 pharmaceutics-13-00165-f009:**
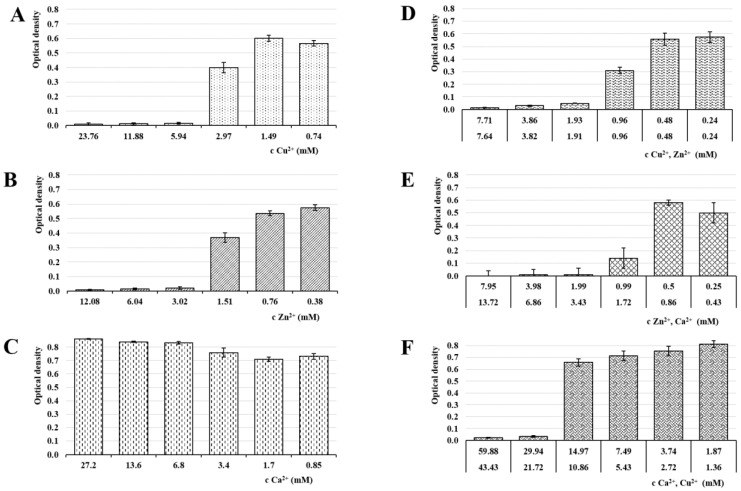
Optical density of *E. coli* in media of individual samples: (A) ALG_Cu, (**B**) ALG_Zn, (**C**) ALG_Ca, (**D**) ALG_Cu:Zn, (**E**) ALG_Zn:Ca and (**F**) ALG_Ca:Cu ions following 24 hrs incubation. The statistically significant effect of ion concentration on optical density of *E. coli* in media was proved by ANOVA (*p* < 0.05 in all cases).

**Figure 10 pharmaceutics-13-00165-f010:**
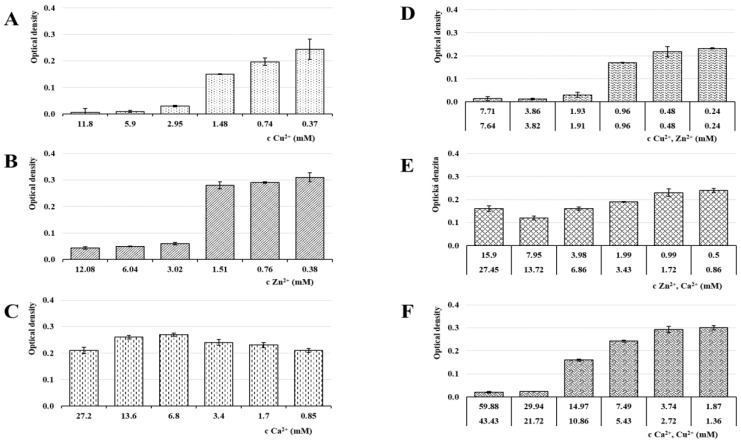
Optical density of *E. faecalis* in media of individual samples: (**A**) ALG_Cu, (**B**) ALG_Zn, (**C**) ALG_Ca, (**D**) ALG_Cu:Zn, (**E**) ALG_Zn:Ca and (**F**) ALG_Ca:Cu ions following 24 hrs incubation. The statistically significant effect of ion concentration on optical density of *E. faecalis* in media was proved by ANOVA (*p* < 0.05 in all cases with the exception of ion concentration effect for Ca^2+^
*p* = 0.097).

**Figure 11 pharmaceutics-13-00165-f011:**
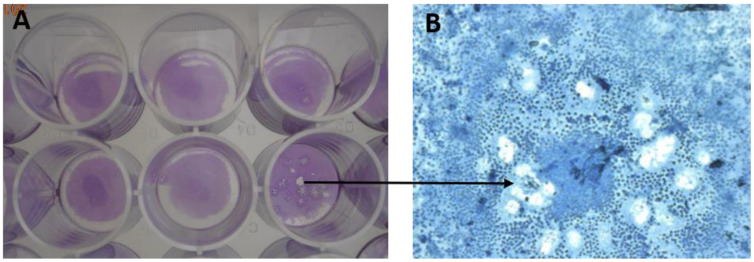
Cytopathogenic effects (CPEs) on cell monolayer stained by crystal violet: (**A**) CPEs in microtiter plates (Nunc) as visible by naked eye; (**B**) CPE imaging using inverse microscope (5 × 10 magnification).

**Table 1 pharmaceutics-13-00165-t001:** Formulation variables of prepared samples.

Sample	NaALG (%)	CuCl_2_ (mol/dm^3^)	ZnCl_2_ (mol/dm^3^)	CaCl_2_ (mol/dm^3^)
ALG_Cu	6	1	-	-
ALG_Zn	6	-	1	-
ALG_Ca	6	-	-	1
ALG_Cu:Zn	6	0.5	0.5	-
ALG_Zn:Ca	6	-	0.5	0.5
ALG_Ca:Cu	6	0.5	-	0.5

**Table 2 pharmaceutics-13-00165-t002:** Yield, equivalent diameter and sphericity factor results.

Sample	Weigh of Dry Samples (g)	Equivalent Diameter (µm)	Sphericity
ALG_Cu	13.99	1455.68 ± 18.71	0.97 ± 0.06
ALG_Zn	14.22	1688.44 ± 12.45	0.90 ± 0.08
ALG_Ca	12.95	1501.50 ± 15.61	0.94 ± 0.07
ALG_Cu:Zn	17.24	1691.77 ± 15.92	0.91 ± 0.07
ALG_Zn:Ca	15.29	1756.31 ± 16.58	0.86 ± 0.04
ALG_Ca:Cu	15.11	1593.31 ± 11.39	0.95 ± 0.07

**Table 3 pharmaceutics-13-00165-t003:** Ion content in individual samples.

Sample	Cu^2+^ (g/kg)	Zn^2+^ (g/kg)	Ca^2+^ (g/kg)
ALG_Cu	205.3 ± 25.1	-	-
ALG_Zn	-	223.8 ± 21.3	-
ALG_Ca	-	-	117.4 ± 15.9
ALG_Cu:Zn	158.7 ± 11.3	87.4 ± 4.4	-
ALG_Zn:Ca	-	123.4 ± 13.7	71.8 ± 5.1
ALG_Ca:Cu	118.9 ± 4.1	-	36.5 ± 2.7

**Table 4 pharmaceutics-13-00165-t004:** The amounts of formed cytopathogenic effects (CPEs) (the average amount of CPEs from the virus control wells was 13 CPEs).

Samples	Ion Concentration (mM)		Average Amounts of CPEs	
			Preventive Efficiency	Therapeutic Efficiency
ALG_Cu	Cu^2+^	0.060	0.8	3.1
ALG_Zn	Zn^2+^	0.600	1.8	5.2
ALG_Ca	Ca^2+^	3.410	8.0	9.2
ALG_Cu:Zn	Cu^2+^	0.035	2.0	7.6
	Zn^2+^	0.034		
ALG_Zn:Ca	Zn^2+^	0.040	3.0	4.8
	Ca^2+^	0.037		
ALG_Ca:Cu	Ca^2+^	0.065	1.0	2.8
	Cu^2+^	0.048		

## Data Availability

Data are contained within the article.

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
