# Peer review of "Assessment of Antimicrobic, Antivirotic and Cytotoxic Potential of Alginate Beads Cross-Linked by Bivalent Ions for Vaginal Administration"

_pharmaceutics, 2021, doi:10.3390/pharmaceutics13020165_

Round 1

Reviewer 1 Report

The manuscript "Antimicrobic and antivirotic effect of alginate beads cross-linked by bivalent ions for vaginal administration" shows that alginate beads stabilized with metallic ions (Ca, Zn, Cu, and combined systems). The study is well organized and present new and exciting results, principally about the cytotoxicity toward Vero cells, antimicrobial, and antiviral properties. However, the manuscript can be improved. The authors can provide EDS spectra by mapping the elements in the beads. Please, see the comments below.

Abstract
Please, provide the full name for the acronyms HHV-1 and TC50.

Introduction

The authors should say the real novelty of the manuscript, discussing and comparing the manuscript proposal with data already reported in the literature. What are the advantages of the proposed materials compared to other alginate-based materials already reported?

Experimental and result sections

What are the alginate molecular mass and ratio between G/M units? Can the authors provide these data?

Table 1. Why were these metallic ions' concentration selected?

I miss some chemical characterization, like FTIR and EDS. The authors can use the X-ray energy dispersive spectrum (EDS) to map the hydrogel structure's metallic ions.
The FTIR can show that the alginate comprises the hydrogel and indicate different absorption bands in the FTIR spectra associated with the alginate-Ca, alginate-Zn, and alginate-Cu.

Cytotoxicity assay
Please, provide more details about the methylene blue staining. Is the AlamarBlue assay or MTT assay? It is not clear what was method used to obtain the cytotoxicity.

Fig. 1. Please, change the legend. The terms Ba and Ca are confusing with the metallic element symbols ascribed to the barium and calcium.

The authors say that mucoadhesive beads were prepared, but they did not show any mucoadhesive assay and findings.

The conclusion is not appropriated. The authors must comment on the manuscript novelty. The authors must comment about the metallic ion concentrations that were cytocompatible to the Vero cells and cytotoxic to the microbial cells and antiviral.

Can these materials be applied

The English language must be checked and improved.

Author Response

Dear reviewer, we thank you for the comments. The list with elaboration of the comments is attached.

Reviewer 2 Report

The manuscript submitted is on alginate beads cross-linked by bivalent ions and its evaluation antimicrobic and antivirotic via vaginal route moderate number of research reports has been published till date on this concept. The research has limited topics bringing up the importance explores the alginate beads cross-linked by bivalent ions and antimicrobic and antivirotic effect via vaginal route with bivalent ions which is newer. The research submitted adds significant information in the respective field. The research manuscript is not well articulated but further needs more information, relevant latest references and modifications are required as per the following recommendations. The authors are even advised strongly for English language revisions as well. These major revisions and additions are required in order the manuscript to be considered for peer review or acceptance or publication further.

  1. Title seems to be very indirect, Suggested to be started as “Assessment or evaluation of alginate beads modified or crosslinked….. Authors are strongly suggested for this change.
  2. Abstract is not structured. Authors need to elevate the modifications were made, characterisation, evaluation data not just the techniques used.
  3. The sentence “significant improvement over Cu2” how much? The abstract lacks upbringing or mentioning the obtained research data in abstract. Abstract are the face and glimpse of the manuscript, needs to be as per publication standards.
  4. The abstract states “demonstrated in previous experiment” sounds meaningless, more information needs to be added or mentioned –
  5. Also at the end of abstract should implicate the importance on how evaluations would be beneficial for pharma industries even in eco-friendly direction.
  6. Authors are suggested to mention the objective and importance of this research in abstract. There is no future studies mentioned in conclusion of abstract. Authors are strongly suggested for the needful.
  7. The sentence in abstract “Moreover, the anti HHV-1 activity was demonstrated, especially in prophylactic use- sounds meanginless, should be repharsed. Performing one experiment for evaluation and claiming is not correct.
  8. “effect” - is not keywords. Authors are strongly suggested for this change.
  9. Introduction should be concise to max 1 ½ page and not more.
  10. Introduction lacks to mention the need for this alginate crosslinked ions formulation how this could overcome the conventional or previously reported formulation for same intended use.
  11. Authors have not used or implemented “Quality by Design” approach to obtain or achieve optimized formulation. Authors need to justify. Also authors should explain how 6 formulation were prepared and on the basis of what the optimized formulation was selected.
  12. The statement “Comparing the swelling charts of Figure 3 and Figure 345 4 shows that at the end of the first hour all swelling profiles are comparable, but while in 346 the other samples swelling increased greatly, samples cross-linked with Cu2+ swelled 347 negligibly in the following hours” needs more justification.
  13. Figure 3 and 4,5,6 lacks errors bars post statistical anlaysis at each time point.
  14. The data needs to be mentioned “The ALG_Cu represented a reference sample, as an output from 458 our previous study”.
  15. The TC50 of Cu2+ through out manuscript is not justified with reefrnece to previously reported studies. Please find the below latest referneces appended, authors are advised to add, cite and discuss the same.
  16. Fig 7 lacks statistical significance ? the authors need to perform ANOVA and highlight the significant groups.
  17. Similarly Fig 8,9,10 lacks statistical significance ? the authors need to perform ANOVA and highlight the significant groups.
  18. Conclusion is short, needs to be expanded mentioning the application of the formulation developed and used- further –as future extension of the work.
  19. Also below are the few latest delivery systems developed and evaluated for similar therapeutic indication, authors are strongly advised to cite and mention in the respective discussion sections. Authors are strongly suggested for these recommendations to be considered for further evaluation.

AAPS PharmSciTech18(4), pp.1343-1354.

AAPS PharmSciTech. 2017 May 1;18(4):1343-54.

In Alginates in Drug Delivery, pp. 129-152. Academic Press, 2020.

International Journal of Biological Macromolecules 163 (2020): 1421-1432.

International journal of biological macromolecules 126 (2019): 359-366.

In Alginates in Drug Delivery, pp. 323-358. Academic Press, 2020.

Biological trace element research148(3), pp.415-419.

AAPS PharmSciTech20(7), p.297.

Journal of Applied Polymer Science113(2), pp.757-766.

Materials Today: Proceedings5(8), pp.16258-16266.

Author Response

(The authors gave the same response as above.)

Round 2

Reviewer 1 Report

After the revision process, the manuscript is suitable for publication.